# Dietary Carbohydrates and Insulin Resistance in Adolescents from Marginalized Areas of Chiapas, México

**DOI:** 10.3390/nu11123066

**Published:** 2019-12-16

**Authors:** Itandehui Castro-Quezada, Elena Flores-Guillén, Pilar E. Núñez-Ortega, César A. Irecta-Nájera, Xariss M. Sánchez-Chino, Orquidia G. Mendez-Flores, Zendy E. Olivo-Vidal, Rosario García-Miranda, Roberto Solís-Hernández, Héctor Ochoa-Díaz-López

**Affiliations:** 1Health Department, El Colegio de la Frontera Sur, Carr. Panamericana y Periférico Sur s/n, Barrio de María Auxiliadora, San Cristóbal de las Casas, Chiapas 29290, Mexico; 2Faculty of Nutrition and Food Science, University of Science and Arts of Chiapas, Libramiento Norte-Poniente 1150, Col. Lajas Maciel, Tuxtla Gutiérrez, Chiapas 29039, Mexico; 3Health Department, El Colegio de la Frontera Sur, Carr. A Reforma Km. 15.5 s/n, RA. Guineo 2da. Sección, Villahermosa, Tabasco 86280, Mexico; 4Cátedra-CONACyT, Health Department, El Colegio de la Frontera Sur, Unidad Villahermosa, Carretera a Reforma Km. 15.5 s/n, RA. Guineo 2da. Sección, Villahermosa, Tabasco 86280, Mexico; 5Cátedra-CONACyT, Health Department, El Colegio de la Frontera Sur, Carr. Panamericana y Periférico Sur s/n, Barrio de María Auxiliadora, San Cristóbal de las Casas, Chiapas 29290, Mexico

**Keywords:** carbohydrates, dietary fiber, sugars, glycemic index, glycemic load, insulin, insulin resistance, adolescents, Chiapas, México

## Abstract

Evidence of the role that dietary carbohydrates (total carbohydrates, dietary fiber, total sugars, dietary glycemic index (GI) and glycemic load (GL)) exerts on insulin levels in adolescents is controversial. Thus, the aim of this study was to assess the association between dietary carbohydrates and insulin resistance in adolescents from Chiapas, México. A cross-sectional study was conducted in 217 adolescents. Sociodemographic, anthropometric, dietary and biochemical data were obtained. Total carbohydrates, dietary fiber, total sugars, dietary GI and GL were calculated from 24 h recalls. Two validated cut-off points for the homeostasis model assessment of insulin resistance (HOMA-IR) were used as surrogates of insulin resistance. Fasting insulin levels ≥ 14.38 μU/mL were considered as abnormal. Multivariate logistic regression models were fitted to assess the association between tertiles of dietary carbohydrates and insulin resistance or hyperinsulinemia. In our study, adolescents with the highest dietary fiber intake had lower odds of HOMA-IR > 2.97 (OR = 0.34; 95% CI: 0.13–0.93) when adjusted for sex, age, body fat percentage and saturated fatty acids intake. No significant associations were found for the rest of the carbohydrate variables. In summary, high-fiber diets reduce the probability of insulin resistance in adolescents from marginalized areas of Chiapas, México.

## 1. Introduction

Insulin resistance is a fundamental component of Type 2 diabetes mellitus (T2DM) etiology and is related to a wide range of diseases such as hypertension, hyperlipidemia and polycystic ovarian disease [1]. Diet is a modifiable factor that can prevent or predispose to insulin resistance. Within the dietary factor, the effect of carbohydrate intake on insulin resistance has been scarcely studied in adolescents. Observational studies have found higher insulin sensitivity in youth with diets high in dietary fiber [2] and whole-grain intake [3]. On the contrary, high total sugar intake has been linked to increased insulin resistance in girls and adolescents [4], although this association remains controversial [5].

The effect of carbohydrates on serum glucose and insulin concentration is mainly determined by the amount of carbohydrates consumed and their absorption rate [6]. In order to include these two determinants of glycemic response, two indices have been developed: glycemic index (GI) [7] and glycemic load (GL) [8]. Diets high in GI and GL have been associated with a higher risk of T2DM in adults [9]. High GI diets during puberty have been associated with increased insulin resistance in adulthood [10]. Conversely, low GI diets can produce significant decreases in the homeostasis model assessment of insulin resistance (HOMA-IR) in children and adolescents [11].

These findings suggest that the quantity, type and quality of carbohydrates during adolescence might affect health status in the short and long term, conditioning the development of chronic degenerative diseases. However, only a few studies have evaluated the effect that dietary carbohydrates exert on insulin resistance in Mexican adolescents [12]. Thus, the aim of the present study was to assess the association between carbohydrate nutrition variables and insulin resistance in adolescents from marginalized areas of Chiapas, México.

## 2. Materials and Methods

### 2.1. Study Design

This cross-sectional study was conducted within the framework of a prospective Cohort Study of newborns from the Tzotzil-Tzeltal and Selva regions of Chiapas, México. A full description of the cohort has been already published [13]. In brief, a cohort study of 2184 newborns from three public hospitals in Chiapas was started in 2003 [14]. The initial assessment included sociodemographic data, anthropometric measurements of the newborn and gynecological history of the mother. In the period from 2017 to 2018, follow-up interviews were applied to a subsample of 303 adolescents randomly selected from the original cohort [13]. Household surveys were performed by trained personnel to collect sociodemographic, clinical and anthropometric data. Dietary intake was assessed using 24 h dietary recalls and food frequency questionnaires. Fasting blood samples were also collected. Written informed consent was obtained from all participants and their parents prior to the interview. This study was conducted in accordance with the Declaration of Helsinki and the protocol was approved by the Research Ethics Committee of El Colegio de la Frontera Sur (CEI-O-076/16).

### 2.2. Study Population

For this analysis, we excluded adolescents with missing values of fasting serum glucose (*n* = 48) and insulin levels (*n* = 3). We used Goldberg’s cut-off2 criterion [15,16] to identify diet reports of poor validity. Basal metabolic rate (BMR) was estimated according to sex, age, weight and height using the Institute of Medicine predictive equations [17]. The reported energy intake (EIrep) as a multiple of BMR (EIrep: BMR) was compared to the physical activity level (PAL) using 95% confidence limits according to Black [15]. The 95% confidence limits of agreement were calculated considering the average within-subject variation in energy intake (23%), between-subject variation in PAL (15%) and within-subject variation in estimated BMR (7.5% for males and 9.3% for females) [15]. Due to the absence of an objective measure of physical activity in our study, a sedentary PAL (1.55) was used for all subjects [15]. Thus, lower and upper cut-off values were defined for males (EIrep: BMR <0.88 and >2.74) and females (EIrep: BMR <0.87 and >2.77). The latter values are similar to those found in a large sample of children and adolescents from the USA [18]. Adolescents with an estimated EIrep: BMR outside the 95% confidence limits were classified as implausible reporters of energy intake (*n* = 35) and were excluded from the analysis. Therefore, the final sample included 217 adolescents (Figure 1).

### 2.3. Dietary Assessment

Trained dietitians administered a single 24 h recall to collect food intake [13]. Support materials were used to assess the portion size consumed (examples of volumes and household measures). Food intake reported in household measures was converted to g/day. Energy (kcal/day) and nutrient intakes (g/day) were calculated by multiplying the portion size of the food consumed (g/day) per grams of nutrient content of the food (g/g). The nutrient data bank was updated using the Nutrient Composition Tables for Mexican foods [19], food composition tables compiled by the National Institute of Public Health of México [20] and the USDA food composition databases [21].

### 2.4. Dietary GI and GL Estimation

We assigned the GI value to each food item reported in the 24 h recalls, according to a standardized methodology [22] using data from Mexican studies [23,24] and the International Tables of GI and GL Values [25]. Dietary GI was calculated as follows [26]: Ʃ [GI * Amount of available carbohydrates of the food (g)]/Ʃ [Amount of available carbohydrates of the food (g)]. Dietary GL was estimated using the following equation: Ʃ [GI * Amount of available carbohydrates of the food (g)]/100.

### 2.5. Sociodemographic Data

A structured questionnaire was administered through interviews to collect information on age, sex, mother’s language (Spanish or Indigenous), mother’s education and family history of diabetes or obesity. Mother’s education was classified into five categories considering the last degree of studies approved (illiterate, elementary school, middle school, high school and bachelor’s degree).

### 2.6. Biochemical Parameters

Blood was drawn from the antecubital vein, after a fasting period of 12 h. Serum glucose was determined by enzymatic photometric tests (Glucose GOD FS, DiaSys Diagnostic Systems GmbH) in an automated analyzer (Vitalab Selectra E, Vitalab Scientific). Serum insulin levels were determined by an immunoassay method (Adaltis Diagnostics) using an Eclectica analyzer (Adaltis Diagnostics). The cut-off point of 14.38 was considered for abnormal insulin levels since it has been proven to have an adequate performance to diagnose metabolic syndrome for a similar population to our sample [27].

The homeostasis model assessment of insulin resistance (HOMA-IR) was calculated using the formula [28]: [fasting insulin (µU/mL)*fasting glucose (mg/dL)]/405. Two validated cut-off points were used to diagnose insulin resistance: HOMA-IR > 3.16 as proposed by Keskin et al. in 2005, which has been widely used in epidemiological studies [29], and HOMA-IR > 2.97 proposed by Piña-Aguero et al. in 2018 as a valid cut-off for insulin resistance in Mexican adolescents [27].

### 2.7. Anthropometric Measurements

Weight (kg) was measured using scales (TANITA Corporation, Tokyo, Japan) and height (cm) was measured using stadiometers (Seca GmbH & co., Hamburg, Germany) with a precision of 100 g and 1 mm, respectively. Body mass index (BMI) was calculated as weight (kg) divided by height squared (m)^2^. BMI z-scores were estimated with the WHO AnthroPlus software [30]. Weight status was categorized as: underweight [(z-score of BMI for age ≤ −2 standard deviations (SD)], normal weight (−1.9 to +1 SD), overweight (>+1 SD) or obese (>+2 SD). Body fat percentage (%BF) was estimated using BIA scales (TANITA Corporation, Tokyo, Japan). Excess of body fat was defined as follows: ≥30% BF for females and ≥25% BF for males, since these thresholds have been associated with alterations in cardiovascular disease risk factors in children and adolescents [31]. Waist circumference was determined with a measuring tape with a precision of 1 mm (Seca GmbH & Co. KG., Hamburg, Germany). Abdominal obesity was defined using the International Diabetes Federation cut-off points for age (Waist circumference ≥ 90 cm for males and ≥80 cm for females) [32]. Furthermore, waist to height ratio (WHtR) was calculated as waist (cm) divided by height (cm) and the cut-off point > 0.5 units was used since it correlates well with metabolic abnormalities in Mexican youth [33]. All measurements were taken wearing light clothes, barefoot, in accordance with the International Society for the Advancement of Kinanthropometry procedures [34].

### 2.8. Statistical Analysis

Carbohydrate, protein, fat, monounsaturated fatty acids (MUFAs), polyunsaturated fatty acids (PUFAs), saturated fatty acids (SFAs), dietary fiber, total sugars and dietary GL intakes were adjusted for total energy intake using the residuals method proposed by Willett [35]. Energy-adjusted carbohydrate intake variables and dietary GI were categorized into tertiles. We performed a descriptive analysis of the sample characteristics according to categories of energy-adjusted dietary fiber intake, using percentages for categorical variables or means and standard deviations (SDs) for continuous variables. Medians and interquartile ranges (IQRs) were calculated for highly skewed biochemical and dietary measurements. Differences across dietary fiber categories were analyzed using ANOVA or Kruskal–Wallis tests for continuous variables, and Chi-square tests for categorical variables.

Multivariate logistic regression models were fitted to estimate Odds Ratios (ORs) with 95% confidence intervals (CIs) to assess the association between carbohydrate intake variables and insulin resistance or hyperinsulinemia. The lowest category of carbohydrate intake variables was used as reference in all the models presented. We fitted four logistic regression models: the first model was unadjusted (crude model); the second model was adjusted for sex and age; the third model was further adjusted for socioeconomic and/or anthropometric variables (geographic area, region, mother’s education level, mother’s language, family history of diabetes or obesity, weight status, waist circumference, %BF or WHtR); the fourth model was further adjusted for dietary intake variables: protein as g/day or % of total energy intake, fat as g/day or % of total energy intake, MUFAs (g/day), PUFAs (g/day), SFAs (g/day) and dietary fiber intakes (g/day) and/or total energy intake (kcal/day). Wald tests and likelihood ratio tests were conducted to select significant covariates. We also performed Hosmer–Lemeshow tests and used individual classification criteria (sensitivity, specificity) to assess the goodness-of-fit of the models. Tests of linear trend across increasing categories of carbohydrate intake variables were conducted by assigning the medians to each category and treating these variables as continuous in the logistic regression models. To assess a possible interaction between carbohydrate intake variables and sex, age, geographic area, region, mother’s education level, mother’s language, family history of diabetes or obesity, weight status, waist circumference, %BF or WHtR; the product terms of the variables were introduced in the models and considered *p* < 0.05 in the likelihood ratio test as statistically significant. All analyses were performed using Stata (Version 15.0, StataCorp, College Station, TX, USA).

## 3. Results

General characteristics of the sample according to energy-adjusted dietary fiber intake are described in Table 1. Study participants with the highest dietary fiber intake were more likely to be males, to live in rural areas, to have a mother who speaks an indigenous language, to have normal weight, to have less body fat percentage, compared to participants with the lowest dietary fiber intake. Subjects in the top category of dietary fiber intake also had higher total carbohydrate intake, lower fat intake (from all types), lower total sugar intake and lower dietary GI than those in the bottom category of fiber intake.

Table 2 shows the results for biochemical measurements and the prevalence of insulin resistance and abnormal fasting insulin levels, according to categories of carbohydrate intake variables. In our study, the prevalence of insulin resistance according to two different cut-off points was: HOMA-IR > 3.16 (Overall: 21%; Females: 25%; Males: 17%) and >2.97 (Overall 23%; Females: 28%; Males: 18%). Similarly, abnormal insulin levels were found in 22% of the sample (Females: 27%; Males: 16%). Fasting serum insulin and HOMA-IR median values were higher for subjects with the lowest dietary fiber intake. On the contrary, adolescents with a high dietary fiber intake showed significantly lower prevalences of insulin resistance (9.7%) and abnormal insulin levels (8.3%) than those with the lowest fiber intake (34.3% and 35.6%, respectively). Fasting insulin levels and HOMA-IR values were lower in adolescents with a moderate sugar intake (second tertile), when compared to high sugar intake group. Nevertheless, a higher prevalence of HOMA-IR > 2.97 and hyperinsulinemia was observed for individuals in the highest sugar intake category. Median fasting glucose in adolescents with high dietary GI was 86 mg/dL, and it was significantly higher than the low GI group.

The ORs and 95% CIs for insulin resistance (HOMA-IR > 3.16) according to dietary carbohydrates (total carbohydrate, dietary fiber, total sugars, dietary GL and dietary GI) are presented in Table 3. We found a statistically significant interaction between sex and total carbohydrate intake; no significant interactions were observed for dietary fiber, total sugars, dietary GI or GL. Female adolescents in the top category of carbohydrate intake had lower odds of insulin resistance than those in the bottom category (Model 1: OR = 0.19; 95% CI: 0.05–0.73; *p* for trend = 0.009). Conversely the direction of the effect changed after the adjustment for dietary fiber and MUFAs intake, showing non-significant higher odds of insulin resistance in girls with high total carbohydrate intake (Model 4: OR = 1.08; 95% CI: 0.19–6.20; *p* for trend = 0.839). Furthermore, high dietary fiber intake was associated with lower odds of HOMA-IR > 3.16 (Model 1: OR = 0.21; 95% CI: 0.08–0.52); *p* for trend = 0.001). However, this association was no longer significant after adjustment for sex, age, BF%, energy-adjusted MUFAs intake and total energy intake (kcal/day) (Table 3). Hosmer–Lemeshow tests and individual classification criteria (sensitivity, specificity) of the models are shown in Appendix A.

Similar results were found when the second cut-off (HOMA-IR > 2.97) for insulin resistance was used (Table 4). Adolescents with high fiber intake had lower odds of insulin resistance (HOMA-IR > 2.97) (Model 1: OR = 0.24; 95% CI: 0.10–0.58); *p* for trend = 0.001), which remained statistically significant after adjustment for sex, age, BF% and energy-adjusted SFAs intake. No statistically significant associations were observed between total carbohydrates, total sugars, dietary GI or dietary GL and HOMA-IR > 2.97. Goodness of fit assessment and quality criteria of the models are shown in Appendix A.

Table 5 presents the ORs and 95% CIs for elevated fasting insulin concentration (≥14.38 μU/mL) according to categories of carbohydrate intake variables. Lower odds of hyperinsulinemia were also observed for subjects with high dietary fiber intake (Model 1: OR = 0.16; 95% CI: 0.06–0.43); *p* for trend <0.001). On the contrary, the probability of elevated fasting insulin concentrations was two times higher for adolescents with high dietary GI, when compared to adolescents with a low dietary GI (Model 1: OR = 2.32; 95% CI: 1.02–5.26); *p* for trend = 0.042). However, this association was attenuated, and statistical significance was lost after adjustment for sex, age, mother’s language, weight status and energy-adjusted dietary fiber intake. Appendix A shows the results for Hosmer–Lemeshow tests and individual classification criteria (sensitivity, specificity) of the models.

## 4. Discussion

The median of HOMA-IR in our study was 1.8 units, which is similar to values previously reported for similar aged adolescents in México [36,37]. The prevalence of insulin resistance in our study (21%–23%) was lower than the prevalence found for overweight/obese adolescents from Tuxtla Gutiérrez, the capital of Chiapas (40%) but higher than normal-weight adolescents in Chiapas (16%) [38]. Another study conducted in 292 adolescents (12–15 years) from Coahuila, México, found a higher proportion (46%) of insulin resistance [37]. Contrast between studies might be explained to weight status, since almost half of the sample in the previous study was classified as overweight or obese, whereas in our study, overweight and obesity prevalence was 27%.

We found that high dietary fiber intake was associated with lower odds of insulin resistance in adolescents from Chiapas, México. This finding adds to the growing evidence of the relevance of dietary fiber for health in adolescence. Previous research in other countries have found similar results. For instance, an investigation carried out in 754 adolescents from Georgia, USA, found that fiber intakes (insoluble and soluble) were inversely associated with fasting insulin and HOMA-IR values [39]. A prospective study of American female adolescents (16–17 years) has also shown that increases in dietary soluble fiber during 3 years were related to reductions in HOMA-IR values and insulin levels [40]. Similarly, it has been demonstrated that a 10 g/MJ increase in dietary fiber reduced more than 1 SD of HOMA-IR among Danish girls [4].

Nevertheless, evidence is not consistent and our results differ from studies conducted in children and adolescents from Canada [41] and eight European cities [42], where no association between dietary fiber intake and insulin levels was found. Differences between studies might be attributed to lower exposure levels—in the Canadian study, fiber intake was around 13 g/day [41]; in the European study, the age range was 12.5–17.5 years and mean dietary fiber consumption was 20 g/day [42]. In México, an investigation carried out in 246 adolescents (12–14 years) from private and public schools located in Tuxtla Gutiérrez, Chiapas, did not find a relationship between carbohydrates or fiber intakes and HOMA-IR [12]. Nevertheless, in Tuxtla Gutiérrez, the mean dietary fiber intake was considerably lower (10 g/day) than in our study (26.6 g/day). The latter figure is similar to dietary fiber intake reported for adolescents from rural population at a national level (28 g/day) [43]. This could be attributed to the sample characteristics, since our research included urban and rural population from marginalized areas, where maize and beans are the two most important staple foods of the diet.

The possible protective role of dietary fiber on insulin resistance could be explained by different mechanisms involving different sites of action [44]. It has been demonstrated that dietary fiber increases the sensitivity of peripheral tissues to physiologic concentrations of insulin [45]. In the small intestine, fiber acts as a mechanical barrier and increases intraluminal viscosity, delays intestinal transit and reduces glucose absorption and insulin secretion. In addition, dietary fiber increases satiety, contributing to disrupt mechanisms leading to obesity. In the large intestine, fiber fermentation results in the production of short-chain fatty acids, which have been associated with the improvement of insulin sensitivity [44]. Also, fermentable fiber intake (oligofructose) may alter microbiota composition, reducing gram-negative bacterial content [46]. The gram-negative bacterial lipopolysaccharides have been associated with increased insulin levels in mice [47]. Therefore, dietary fiber could reduce insulin resistance by relieving endotoxemia [44,47].

In our study, no significant associations were observed for the rest of the carbohydrate variables. Likewise, other studies conducted in youth have not found a link between total carbohydrate intake [10], total sugar intake [10], dietary GI [48] or dietary GL [10,48] and insulin resistance. However, experimental evidence of such associations is controversial. It has been demonstrated that a high-carbohydrate/low-fat diet increased insulin sensitivity in normal-weight adolescents [49], but in obese subjects (13–17 years) the same diet increased insulin secretion without improvements on insulin sensitivity [50].

Regarding total sugar intake, a clinical trial conducted in weight-stable, physically active adolescents showed that a moderate intake of fructose and glucose beverages for a period of 2 weeks had no effect on insulin levels [51]. In contrast, a short-term (9 days) restriction of sugar and fructose (within an isocaloric diet) has shown to be effective in decreasing liver fat, visceral fat and improving insulin dynamics (sensitivity, secretion, and clearance) in obese children [52]. A prospective study conducted in 120 overweight Latino children in California showed that high sugar consumption was correlated to lower acute insulin responses and a reduced disposition index (index of β-cell function) [48]. The evidence suggests that the detrimental effect of high sugar intake on insulin responses could be more pronounced in subjects with overweight or obesity. However, in our study, no significant interaction was found for total sugar intake and weight status. Our results showed higher odds for insulin resistance for subjects in the highest tertile of total sugar intake, although it did not reach statistical significance. Furthermore, there is evidence from clinical trials that low-GI/low GL diets reduce HOMA index in children and adolescents [11]. Our results showed a non-significant trend of higher odds of insulin resistance for the highest categories of dietary GI and GL in the multivariate model.

The inconsistency of results could be explained by some methodological limitations of our study. The use of a single administration of a 24 h recall did not allow us to account for day-to-day variations of habitual dietary intake. As the precision of the method improves with the number of recalls administered in the same study subject [53], it would have been preferable to apply the recall at least two or three times. However, the fieldwork was conducted in different municipalities with difficult access and we were not able to repeat the dietary assessment. The 24 h dietary recall has other disadvantages, the reported data relies on the study subject’s recent memory and depends on the interviewer’s capacity for describing ingredients, food preparation and dishes [53]. However, to warrant the accuracy of dietary data, a group of dietitians was specifically trained for this study, the 24 h recall method was applied using tablets with data collection and codification in real-time [13]. Furthermore, we assessed the plausibility of caloric intake by age, weight and height and implausible reporters of energy intake were excluded from the analysis. Misreporting may alter diet and disease associations [54]. Previous studies have demonstrated that excluding implausible reporters yielded results that were more consistent with expectations, when the association between dietary factors and weight status was evaluated [55]. Another limitation in our study is that physical activity and pubertal stage data were not available, thus, we were not able to adjust the models for these variables as potential confounders. Recent studies provide evidence that regular physical activity lowers the risk of insulin resistance and improves insulin sensitivity when individuals comply with physical activity guidelines [56]. As regards to the pubertal stage, evidence shows that transition from Tanner stage I to Tanner stage III is related to reduced insulin sensitivity, with increments in fasting glucose and insulin levels [57]. Nevertheless, models were adjusted for sex and age to reduce this source of error. Another potential weakness of our study is that GI values were assigned to local foods using data from previous studies of Mexican traditional dishes [23] and the International tables of GI and GL values [25], where most values were published for Australian or American foods. This might produce misclassification of foods to GI and GL categories, and probably causing bias towards the null. Finally, the cross-sectional design of our study allowed us to assess only a statistical association. Thus, causality cannot be inferred from the results obtained from the analysis.

A strength of this study is the assignment of GI values through a systematic protocol published for 24 h recalls [22], and for most of the foods with available carbohydrate content (59.6%), the GI values of a closely related food or an exact match in the GI databases were assigned. Also, we used two cut-off points for HOMA-IR as surrogates of insulin resistance, which were previously validated for adolescents [29] and have been demonstrated as a valid predictor of Metabolic Syndrome in a similar population group [27]. Furthermore, our study was conducted in a sample of adolescents from marginalized communities in the South of México, comprising a high proportion of indigenous population (44% of the adolescents’ mothers spoke Mayan language) where few studies have evaluated the association between diet and insulin resistance.

In conclusion, this study provides evidence that high dietary fiber intake is associated with lower odds of insulin resistance in adolescents from Chiapas, México. However, further longitudinal research on carbohydrate nutrition and insulin resistance is required to effectively develop and promote dietary recommendations that may help to prevent Type 2 diabetes incidence and other chronic diseases among this vulnerable population.

## Figures and Tables

**Figure 1 nutrients-11-03066-f001:**
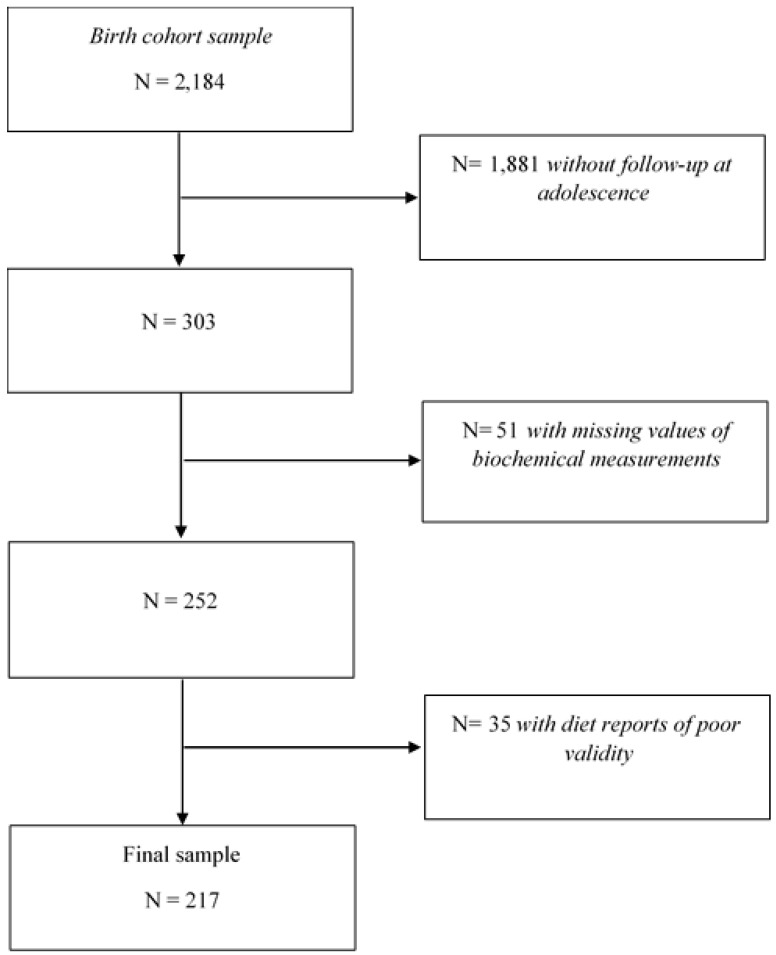
Flow chart showing participant selection.

**Table 1 nutrients-11-03066-t001:** General characteristics of the sample according to categories of dietary fiber intake in a sample of adolescents from Chiapas, México ^a^.

Variables	Energy-Adjusted Dietary Fiber Intake				*p*-Value ^b^
	T1	T2	T3	Total	
	*n* = 73	*n* = 72	*n* = 72	*n* = 217	
**Demographic Characteristics**					
Sex (% female)	57.5	52.8	36.1	48.9	0.026
Age (years) ^c^	14.1 (13.9–14.2)	14.1 (14.0–14.3)	14.2 (14.0–14.4)	14.1 (14.0–14.3)	0.004 ^d^
Geographic area (%)					<0.001
Urban	93.2	81.9	51.4	75.6	
Rural	6.9	18.1	48.6	24.4	
Region (%)					0.034
Altos	74.0	72.2	55.6	67.3	
Selva	26.0	27.8	44.4	32.7	
Mother’s education level (%)					<0.001
Illiterate	21.7	35.2	55.7	37.6	
Elementary school	14.5	25.4	21.4	20.5	
Middle school	39.1	25.4	20.0	28.1	
High school	13.0	9.9	1.4	8.1	
Bachelor’s degree	11.6	4.2	1.4	5.7	
Mother’s language (%)					<0.001
Spanish	76.7	50.0	30.6	52.5	
Indigenous (Mayan)	17.8	48.6	66.7	44.2	
No data available	5.5	1.4	2.8	3.2	
Family history of diabetes (%)	54.8	45.8	37.5	46.1	0.153
Family history of obesity (%)	27.8	20.8	19.4	22.7	0.441
Weight status (%)					0.029
Underweight	1.4	1.4	0.0	0.9	
Normal weight	58.9	75.0	83.3	72.4	
Overweight	31.5	18.1	16.7	22.1	
Obesity	8.2	5.6	0.0	4.6	
Waist circumference (cm)	73.0 (68.0–79.0)	71.0 (66.0–75.3)	72.0 (69.0–75.0)	72.0 (68.0–77.0)	0.160 ^d^
Abdominal obesity (%)	19.2	12.5	9.7	13.8	0.237
% Body fat	24.8 (18.9–30.2)	24.3 (14.2–29.3)	19.8 (14.1–24.9)	22.8 (15.0–28.4)	0.002 ^d^
Body fat excess (%)	31.5	22.2	8.3	20.7	0.002
WHtR (units)	0.5 (0.4–0.5)	0.5 (0.4–0.5)	0.5 (0.5–0.5)	0.5 (0.4–0.5)	0.116 ^d^
High WHtR (>0.5 units) (%)	27.4	25.0	27.8	26.7	0.920
**Dietary Intake**					
Energy intake (kcal/day)	2139 (1970–2566)	2119 (1672–2610)	2069 (1759–2668)	2128 (1782–2588)	0.454 ^d^
Total carbohydrates (g/day) ^e^	307.8 (45.3)	329.3 (40.5)	368.8 (42.5)	335.2 (49.6)	<0.001
Total carbohydrates (% energy)	55.6 (8.7)	59.7 (7.8)	67.2 (8.1)	60.8 (9.5)	<0.001
Protein (g/day) ^c,e^	64.9 (51.1–78.5)	71.9 (61.5–86.0)	69.7 (60.6–79.9)	69.5 (58.7–82.7)	0.089 ^d^
Protein (% energy)	11.7 (9.4–15.0)	13.1 (11.1–15.0)	12.8 (10.8–14.4)	12.6 (10.6–14.6)	0.117 ^d^
Total fat (g/day) ^e^	81.2 (19.4)	69.0 (14.2)	53.3 (18.1)	67.9 (20.8)	<0.001
Total fat (% energy)	33.0 (8.2)	28.0 (6.1)	21.0 (7.8)	27.4 (8.9)	<0.001
MUFAs (g/day) ^e^	25.9 (7.9)	22.6 (7.1)	16.1 (7.4)	21.5 (8.5)	<0.001
PUFAs (g/day) ^c,e^	14.8 (10.4–19.7)	11.5 (9.4–15.0)	10.7 (6.9–14.7)	11.9 (8.9–17.2)	<0.001 ^d^
SFAs (g/day) ^e^	27.9 (9.2)	22.4 (6.6)	16.7 (8.8)	22.4 (9.4)	<0.001
Dietary fiber (g/day) ^c,e^	18.7 (16.2–20.3)	26.6 (25.0–29.7)	38.1 (34.3–43.8)	26.6 (20.3–34.3)	<0.001 ^d^
Total sugars (g/day) ^c,e^	110.8 (69.0–142.0)	82.7 (66.6–95.0)	60.1 (37.3–76.3)	79.1 (55.4–103.6)	<0.001 ^d^
Dietary GI (g/day) ^c^	53.5 (50.5–56.8)	51.5 (49.1–54.4)	48.3 (46.2–51.4)	51.2 (47.7–54.3)	<0.001 ^d^
Dietary GL (g/day) ^e^	162.6 (29.7)	169.5 (29.0)	178.2 (25.0)	170.1 (28.6)	0.004

^a^ Means (SD) or percentages are shown. ^b^ Differences across dietary fiber categories were analyzed using ANOVA for continuous variables and Chi-square tests for categorical variables. ^c^ Medians (Interquartile range). ^d^ Kruskal–Wallis test. ^e^ Energy-adjusted using the residuals method. T, tertiles; MUFAs, monounsaturated fatty acids; PUFAs, polyunsaturated fatty acids; SFAs, saturated fatty acids; WHtR, waist to height ratio; GI, glycemic index, GL, glycemic load.

**Table 2 nutrients-11-03066-t002:** Biochemical measurements and the prevalence of insulin resistance and abnormal fasting insulin levels according to categories of carbohydrate intake variables in adolescents from Chiapas, México.

Variables	n	Parameters					
		Fasting Serum Glucose (mg/dL) ^a^	Fasting Serum Insulin (μU/mL) ^a^	HOMA-IR (Units) ^a^	HOMA-IR > 3.16 (%)	HOMA-IR > 2.97 (%)	Fasting Serum Insulin ≥ 14.38 μU/mL (%)
Total Sample	217	83.0 (80.0–88.0)	9.1 (1.0–13.6)	1.8 (0.2–2.8)	21.2	23.0	21.7
Total Carbohydrates ^b^							
T1	73	84.0 (80.0–87.0)	9.7 (1.0–15.8)	2.1 (0.2–3.4)	27.4	27.4	27.4
T2	72	83.0 (77.8–87.5)	6.1 (1.0–13.1)	1.3 (0.2–2.8)	22.2	23.6	23.6
T3	72	84.0 (79.5–89.0)	9.3 (4.0–12.5)	1.9 (0.8–2.6)	13.9	18.1	13.9
*p*-value ^c^		0.675	0.189	0.194	0.133	0.406	0.126
Dietary Fiber ^b^							
T1	73	85.0 (80.0–89.0)	10.1 (1.0–18.7)	2.2 (0.2–4.4)	34.3	34.3	35.6
T2	72	84.0 (80.0–88.5)	5.6 (1.0–12.7)	1.1 (0.2–2.8)	19.4	23.6	20.8
T3	72	82.3 (78.0–87.0)	8.7 (3.2–12.2)	1.8 (0.6–2.5)	9.7	11.1	8.3
*p*-value ^c^		0.191	0.045	0.046	0.001	0.004	<0.001
Total Sugars ^b^							
T1	73	83.0 (81.0–88.0)	10.4 (1.0–13.6)	2.1 (0.2–2.8)	21.9	23.3	21.9
T2	72	83.0 (78.5–87.0)	5.6 (1.0–10.6)	1.1 (0.2–2.2)	13.9	13.9	12.5
T3	72	83.0 (79.0–89.5)	9.6 (1.0–17.6)	2.1 (0.2–3.6)	27.8	31.9	30.6
*p*-value ^c^		0.543	0.009	0.008	0.123	0.036	0.031
Dietary GI							
T1	73	82.0 (79.0–85.0)	9.1 (1.0–12.3)	1.8 (0.2–2.5)	15.1	15.1	15.1
T2	72	84.0 (77.0–89.5)	7.7 (1.0–13.4)	1.5 (0.2–2.7)	20.8	23.6	20.8
T3	72	86.0 (82.0–90.5)	9.4 (1.0–16.6)	1.9 (0.2–3.9)	27.8	30.6	29.2
*p*-value ^c^		0.001	0.488	0.336	0.173	0.085	0.117
Dietary GL ^b^							
T1	73	83.0 (80.0–86.0)	9.1 (1.0–14.8)	1.8 (0.2–3.4)	26.0	26.0	26.0
T2	72	82.0 (78.5–87.0)	9.0 (1.0–12.8)	1.9 (0.2–2.6)	19.4	19.4	20.8
T3	72	86.0 (80.0–90.0)	9.2 (1.0–13.3)	1.8 (0.2–2.7)	18.1	23.6	18.1
*p*-value ^c^		0.193	0.929	0.926	0.454	0.636	0.496

^a^ Medians (interquartile range) are shown. ^b^ Energy-adjusted using the residuals method. ^c^ Differences across categories were analyzed using Kruskal–Wallis test for continuous variables and Chi-square tests for categorical variables. HOMA-IR, homeostasis model assessment of insulin resistance.

**Table 3 nutrients-11-03066-t003:** Associations between categories of carbohydrate intake variables and insulin resistance (HOMA-IR > 3.16) in adolescents from Chiapas, México.

Factor Variables			Insulin Resistance (HOMA-IR > 3.16)			
			OR (95% CI)	OR (95% CI)	OR (95% CI)	OR (95% CI)
Total Carbohydrates ^a,b^	n	Median	Model 1	Model 2 ^c^	Model 3 ^d^	Model 4 ^e^
Males						
T1	29	286.2	1 (Ref.)	1 (Ref.)	1 (Ref.)	1 (Ref.)
T2	38	335.4	2.69 (0.66–11.01)	2.92 (0.70–12.16)	2.96 (0.69–12.67)	5.04 (0.92–27.52)
T3	44	387.9	1.64 (0.39–6.94)	1.92 (0.44–8.35)	1.97 (0.44–8.80)	4.76 (0.59–38.27)
*p-trend*			0.667	0.510	0.498	0.181
Females						
T1	44	290.9	1 (Ref.)	1 (Ref.)	1 (Ref.)	1 (Ref.)
T2	34	337.3	0.41 (0.15–1.15)	0.43 (0.15–1.24)	0.44 (0.15–1.26)	1.44 (0.37–5.52)
T3	28	378.1	0.19 (0.05–0.73)	0.20 (0.05–0.79)	0.19 (0.05–0.75)	1.08 (0.19–6.20)
*p-trend*			0.009	0.013	0.011	0.839
Dietary Fiber ^a^			Model 1	Model 2 ^f^	Model 3 ^g^	Model 4 ^h^
T1	73	18.7	1 (Ref.)	1 (Ref.)	1 (Ref.)	1 (Ref.)
T2	72	26.6	0.46 (0.22–0.99)	0.51 (0.23–1.10)	0.51 (0.24–1.11)	0.61 (0.28–1.37)
T3	72	38.1	0.21 (0.08–0.52)	0.26 (0.10–0.68)	0.28 (0.11–0.72)	0.46 (0.16–1.31)
*p-trend*			0.001	0.005	0.007	0.129
Total Sugars ^a^			Model 1	Model 2 ^f^	Model 3 ^g^	Model 4 ^i^
T1	73	46.0	1 (Ref.)	1 (Ref.)	1 (Ref.)	1 (Ref.)
T2	72	79.5	0.57 (0.24–1.37)	0.53 (0.22–1.29)	0.56 (0.23–1.36)	0.63 (0.25–1.60)
T3	72	120.4	1.37 (0.64–2.92)	1.10 (0.50–2.42)	1.14 (0.51–2.53)	1.29 (0.56–2.96)
*p-trend*			0.337	0.706	0.640	0.442
Dietary GI			Model 1	Model 2 ^f^	Model 3 ^j^	Model 4 ^k^
T1	73	46.5	1 (Ref.)	1 (Ref.)	1 (Ref.)	1 (Ref.)
T2	72	51.3	1.48 (0.63–3.50)	1.25 (0.52–3.01)	1.21 (0.50–2.95)	1.43 (0.57–3.58)
T3	72	55.7	2.17 (0.95–4.94)	1.68 (0.71–3.98)	1.49 (0.62–3.58)	1.43 (0.59–3.48)
*p-trend*			0.064	0.229	0.365	0.448
Dietary GL ^a^			Model 1	Model 2 ^f^	Model 3 ^j^	Model 4 ^l^
T1	73	144.1	1 (Ref.)	1 (Ref.)	1 (Ref.)	1 (Ref.)
T2	72	172.9	0.69 (0.31–1.50)	0.68 (0.31–1.52)	0.84 (0.37–1.90)	1.22 (0.50–2.94)
T3	72	198.3	0.63 (0.28–1.39)	0.71 (0.31–1.60)	0.79 (0.34–1.83)	1.78 (0.64–4.96)
*p-trend*			0.235	0.370	0.567	0.282

^a^ Energy-adjusted using the residuals method. ^b^ Interaction term *p* < 0.05. ^c^ Adjusted for age (years). ^d^ Adjusted for age (years) and BF% (normal/high). ^e^ Adjusted for age (years), BF% (normal/high), energy-adjusted dietary fiber intake (g/day) and energy-adjusted MUFAs intake (g/day). ^f^ Adjusted for sex (males/females) and age (years). ^g^ Adjusted for sex (males/females), age (years) and BF% (normal/high). ^h^ Adjusted for sex (males/females), age (years), BF% (normal/high), energy-adjusted MUFAs intake (g/day) and total energy intake (kcal/day). ^i^ Adjusted for sex (males/females), age (years), BF% (normal/high) and energy-adjusted MUFAs intake (g/day). ^j^ Adjusted for sex (males/females), age (years), geographic area (rural/urban) and BF% (normal/high). ^k^ Adjusted for sex (males/females), age (years), geographic area (rural/urban), BF% (normal/high), energy-adjusted MUFAs intake (g/day) and total energy intake (kcal/day). ^l^ Adjusted for sex (males/females), age (years), geographic area (rural/urban), BF% (normal/high) and energy-adjusted MUFAs intake (g/day). BF%, body fat percentage.

**Table 4 nutrients-11-03066-t004:** Association between insulin resistance (HOMA-IR > 2.97) and categories of carbohydrate intake variables in adolescents from Chiapas, México.

Factor Variables			Insulin Resistance (HOMA-IR > 2.97)			
			OR (95% CI)	OR (95% CI)	OR (95% CI)	OR (95% CI)
Total Carbohydrates ^a^	n	Median	Model 1	Model 2 ^b^	Model 3 ^c^	Model 4 ^d^
T1	73	290.2	1 (Ref.)	1 (Ref.)	1 (Ref.)	1 (Ref.)
T2	72	335.9	0.82 (0.39–1.73)	0.92 (0.43–1.98)	0.93 (0.43–1.99)	1.09 (0.50–2.40)
T3	72	383.9	0.58 (0.26–1.29)	0.72 (0.32–1.62)	0.70 (0.31–1.59)	1.15 (0.45–2.93)
*p-trend*			0.183	0.429	0.398	0.760
Dietary Fiber ^a^			Model 1	Model 2 ^b^	Model 3 ^c^	Model 4 ^e^
T1	73	18.7	1 (Ref.)	1 (Ref.)	1 (Ref.)	1 (Ref.)
T2	72	26.6	0.59 (0.29–1.23)	0.64 (0.30–1.34)	0.65 (0.31–1.36)	0.69 (0.32–1.50)
T3	72	38.1	0.24 (0.10–0.58)	0.29 (0.12–0.72)	0.30 (0.12–0.76)	0.34 (0.13–0.93)
*p-trend*			0.001	0.007	0.010	0.035
Total Sugars ^a^			Model 1	Model 2 ^b^	Model 3 ^c^	Model 4 ^f^
T1	73	46.0	1 (Ref.)	1 (Ref.)	1 (Ref.)	1 (Ref.)
T2	72	79.5	0.53 (0.22–1.26)	0.49 (0.21–1.18)	0.51 (0.21–1.23)	0.54 (0.22–1.32)
T3	72	120.4	1.55 (0.74–3.22)	1.27 (0.59–2.73)	1.31 (0.61–2.82)	1.42 (0.65–3.10)
*p-trend*			0.176	0.413	0.377	0.278
Dietary GI			Model 1	Model 2 ^b^	Model 3 ^c^	Model 4 ^g^
T1	73	46.5	1 (Ref.)	1 (Ref.)	1 (Ref.)	1 (Ref.)
T2	72	51.3	1.74 (0.75–4.04)	1.54 (0.65–3.64)	1.53 (0.65–3.61)	1.70 (0.70–4.09)
T3	72	55.7	2.48 (1.10–5.60)	2.09 (0.90–4.87)	2.00 (0.85–4.70)	1.83 (0.77–4.35)
*p-trend*			0.028	0.088	0.334	0.240
Dietary GL ^a^			Model 1	Model 2 ^b^	Model 3 ^h^	Model 4 ^i^
T1	73	144.1	1 (Ref.)	1 (Ref.)	1 (Ref.)	1 (Ref.)
T2	72	172.9	0.69 (0.31–1.50)	0.68 (0.31–1.52)	0.68 (0.31–1.51)	0.79 (0.35–1.79)
T3	72	198.3	0.88 (0.41–1.87)	1.02 (0.47–2.21)	0.99 (0.45–2.15)	1.23 (0.55–2.79)
*p-trend*			0.704	0.968	0.907	0.669

^a^ Energy-adjusted using the residuals method. ^b^ Adjusted for sex (males/females) and age (years). ^c^ Adjusted for sex (males/females), age (years) and BF% (normal/high). ^d^ Adjusted for sex (males/females), age (years), BF% (normal/high) and energy-adjusted dietary fiber intake (g/day). ^e^ Adjusted for sex (males/females), age (years), BF% (normal/high) and energy-adjusted SFAs intake (g/day). ^f^ Adjusted for sex (males/females), age (years), BF% (normal/high) and lipids (% energy). ^g^ Adjusted for sex (males/females), age (years), BF% (normal/high) and energy-adjusted MUFAs intake (g/day). ^h^ Adjusted for sex (males/females), age (years), waist circumference (normal/high). ^h^ Adjusted for sex (males/females), age (years), waist circumference (normal/high) and energy-adjusted dietary fiber intake (g/day).

**Table 5 nutrients-11-03066-t005:** Associations between categories of carbohydrate intake variables and elevated fasting insulin concentration (≥14.38 μU/mL) in adolescents from Chiapas, México.

Factor Variables			Elevated Fasting Insulin Concentration (≥14.38 μU/mL)			
			OR (95% CI)	OR (95% CI)	OR (95% CI)	OR (95% CI)
Total Carbohydrates ^a^	n	Median	Model 1	Model 2 ^b^	Model 3 ^c^	Model 4 ^d^
T1	73	290.2	1 (Ref.)	1 (Ref.)	1 (Ref.)	1 (Ref.)
T2	72	335.9	0.82 (0.39–1.73)	0.95 (0.44–2.05)	0.93 (0.42–2.04)	1.66 (0.67–4.12)
T3	72	383.9	0.43 (0.18–0.99)	0.54 (0.23–1.30)	0.52 (0.22–1.26)	1.42 (0.45–4.46)
*p-trend*			0.050	0.184	0.161	0.528
Dietary Fiber ^a^			Model 1	Model 2 ^b^	Model 3 ^c^	Model 4 ^e^
T1	73	18.7	1 (Ref.)	1 (Ref.)	1 (Ref.)	1 (Ref.)
T2	72	26.6	0.48 (0.23–1.00)	0.53 (0.25–1.13)	0.56 (0.26–1.21)	0.65 (0.29–1.42)
T3	72	38.1	0.16 (0.06–0.43)	0.21 (0.08–0.57)	0.24 (0.09–0.64)	0.34 (0.12–1.00)
p-trend			<0.001	0.002	0.004	0.047
Total Sugars ^a^			Model 1	Model 2 ^b^	Model 3 ^c^	Model 4 ^d^
T1	73	46	1 (Ref.)	1 (Ref.)	1 (Ref.)	1 (Ref.)
T2	72	79.5	0.51 (0.21–1.24)	0.46 (0.18–1.14)	0.49 (0.20–1.24)	0.54 (0.21–1.40)
T3	72	120.4	1.57 (0.74–3.31)	1.21 (0.55–2.65)	1.25 (0.56–2.78)	1.40 (0.62–3.19)
*p-trend*			0.168	0.492	0.446	0.301
Dietary GI			Model 1	Model 2 ^b^	Model 3 ^f^	Model 4 ^g^
T1	73	46.5	1 (Ref.)	1 (Ref.)	1 (Ref.)	1 (Ref.)
T2	72	51.3	1.48 (0.63–3.50)	1.22 (0.50–2.95)	1.18 (0.48–2.91)	1.05 (0.42–2.66)
T3	72	55.7	2.32 (1.02–5.26)	1.76 (0.74–4.15)	1.47 (0.61–3.57)	1.10 (0.43–2.78)
*p-trend*			0.042	0.189	0.382	0.842
Dietary GL ^a^			Model 1	Model 2 ^b^	Model 3 ^f^	Model 4 ^h^
T1	73	144.1	1 (Ref.)	1 (Ref.)	1 (Ref.)	1 (Ref.)
T2	72	172.9	0.75 (0.35–1.62)	0.74 (0.34–1.64)	0.88 (0.39–2.00)	0.96 (0.42–2.19)
T3	72	198.3	0.63 (0.28–1.39)	0.72 (0.32–1.65)	0.82 (0.35–1.92)	0.93 (0.39–2.23)
*p-trend*			0.242	0.417	0.642	0.875

^a^ Energy-adjusted using the residuals method. ^b^ Adjusted for sex (males/females) and age (years). ^c^ Adjusted for sex (males/females), age (years) and BMI (normal/overweight or obesity). ^d^ Adjusted for sex (males/females), age (years), BMI (normal/overweight or obesity) and energy-adjusted MUFAs intake (g/day). ^e^ Adjusted for sex (males/females), age (years), BMI (normal/overweight or obesity), energy-adjusted MUFAs intake (g/day) and total energy intake (kcal/day). ^f^ Adjusted for sex (males/females), age (years), Mother’s language (Spanish/Indigenous) and BMI (normal/overweight or obesity). ^g^ Adjusted for sex (males/females), age (years), mother’s language (Spanish/Indigenous), BMI (normal/overweight or obesity) and energy-adjusted dietary fiber intake (g/day). ^h^ Adjusted for sex (males/females), age (years), Mother’s language (Spanish/Indigenous), BMI (normal/overweight or obesity) and energy-adjusted dietary fiber intake (g/day) and total energy intake (kcal/day). BMI, body mass index.

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
