# Peer review of "Dietary Carbohydrates and Insulin Resistance in Adolescents from Marginalized Areas of Chiapas, México"

_nutrients, 2019, doi:10.3390/nu11123066_

Round 1

Reviewer 1 Report

This study observed that high fiber diets reduce the probability of insulin resistance in adolescents from marginalized areas of Chiapas, Mexico. This is a very interesting and important research topic, but unfortunately there are some things to check in the statistical and analytical evaluation section of this study. I would suggest the author revise some places to help readers’ understand.

#major points

1) Line 163 states the following, but the results do not appear to be mentioned. Please describe additionally. “We also performed Hosmer–Lemeshow tests and used individual classification criteria (sensitivity, specificity) to assess the goodness-of-fit of the models.”

2) In the case of macronutrients, % of energy intake from nutrients as well as absolute amount of nutrient intake is known to be an important indicator. I wonder if you have analyzed this part too. I would recommend mentioning the results of further analysis. In the same vein, it may be necessary to add energy intakes to the adjustment variable to determine whether these results are due to high energy intakes.

3) Have you analyzed quartile results or quintile results in addition to the OR group results?

4) Lines 206-211 describe interactions between men and women on total carbohydrate intake. Did you analyze the sexes for all other nutrients (fiber, total sugar, GI, GL) that you did not show? If so, please describe further.

5) Also, the authors write: “Conversely the direction of the effect changed after the adjustment for dietary fiber and MUFA intake, showing non-significant higher odds of insulin resistance in girls with high total carbohydrates intake” Do you think that the cause of the change in direction is due to dietary fiber intake? So what is the significance of dietary fiber and MUFA in this model? I wonder what happens when dietary fiber is only included as a calibration variable and then MUFA is additionally included.

Author Response

Please find attached in the PDF file, the answers to each Reviewers´ comments.

Reviewer 2 Report

The paper is interesting and well written. Just a comment... I suggest to repeat the analyses by using a binomial regression model instead of a logistic regression because in a cross-sectional study we might be interested not to estimate the risk of insulin-resistance but the prevalence.

Author Response

(The authors gave the same response as above.)

Reviewer 3 Report

Manuscript by Castro-Quezada et al. presents the association between carbohydrate nutrition and insulin resistance in adolescents from Chiapas in México. The manuscript has a poorly defined goal and research hypotheses, the analysis of the presented results due to the poor range of parameters does not allow the enrichment of existing knowledge, but only to present already known aspects in the narrow population of Mexico. The manuscript is descriptive in nature, there is no mechanism-based explanation.

Author Response

(The authors gave the same response as above.)

Round 2

Reviewer 1 Report

The author responded in good faith to the reviewer's comments.

Reviewer 3 Report

Explanations from the Authors are sufficient.